# MST3 Involvement in Na^+^ and K^+^ Homeostasis with Increasing Dietary Potassium Intake

**DOI:** 10.3390/ijms22030999

**Published:** 2021-01-20

**Authors:** Chee-Hong Chan, Sheng-Nan Wu, Bo-Ying Bao, Houng-Wei Li, Te-Ling Lu

**Affiliations:** 1Department of Nephrology, Chang Bing Show Chwan Memorial Hospital, Lukang, Changhua 505, Taiwan; cheehong.chan@gmail.com; 2Department of Physiology, National Cheng Kung University Medical College, Tainan 70101, Taiwan; snwu@mail.ncku.edu.tw; 3School of Pharmacy, China Medical University, Taichung 406040, Taiwan; bao@mail.cmu.edu.tw; 4Department of Nursing, Asia University, Taichung 41354, Taiwan; 5Department of Medical Laboratory Science and Biotechnology, China Medical University, Taichung 406040, Taiwan; albertli489@gmail.com

**Keywords:** MST3, STK24, high potassium, ENaC, NKCC2, SPAK, OSR1, WNK4

## Abstract

K^+^ loading inhibits NKCC2 (Na-K-Cl cotransporter) and NCC (Na-Cl cotransporter) in the early distal tubules, resulting in Na^+^ delivery to the late distal convoluted tubules (DCTs). In the DCTs, Na^+^ entry through ENaC (epithelial Na channel) drives K^+^ secretion through ROMK (renal outer medullary potassium channel). WNK4 (with-no-lysine 4) regulates the NCC/NKCC2 through SAPK (Ste20-related proline-alanine-rich kinase)/OSR1 (oxidative stress responsive). K^+^ loading increases intracellular Cl^−^, which binds to the WNK4, thereby inhibiting autophosphorylation and downstream signals. Acute K^+^ loading-deactivated NCC was not observed in Cl^−^-insensitive WNK4 mice, indicating that WNK4 was involved in K^+^ loading-inhibited NCC activity. However, chronic K^+^ loading deactivated NCC in Cl^−^-insensitive WNK4 mice, indicating that other mechanisms may be involved. We previously reported that mammalian Ste20-like protein kinase 3 (MST3/STK24) was expressed mainly in the medullary TAL (thick ascending tubule) and at lower levels in the DCTs. MST3*^−/−^* mice exhibited higher ENaC activity, causing hypernatremia and hypertension. To investigate MST3 function in maintaining Na^+^/K^+^ homeostasis in kidneys, mice were fed diets containing various concentrations of Na^+^ and K^+^. The 2% KCl diets induced less MST3 expression in MST3*^−/−^* mice than that in wild-type (WT) mice. The MST3*^−/−^* mice had higher WNK4, NKCC2-S130 phosphorylation, and ENaC expression, resulting in lower urinary Na^+^ and K^+^ excretion than those of WT mice. Lower urinary Na^+^ excretion was associated with elevated plasma [Na^+^] and hypertension. These results suggest that MST3 maintains Na^+^/K^+^ homeostasis in response to K^+^ loading by regulation of WNK4 expression and NKCC2 and ENaC activity.

## 1. Introduction

An increase in dietary K^+^ intake stimulates aldosterone release, which stimulates renal K^+^ secretion, and does not influence Na^+^ retention. Several Na^+^ and K^+^ channels coordinate to maintain K^+^ secretion without Na^+^ retention. The model of the process suggests that K^+^ loading inhibits NKCC2 in the loop of Henle [1,2] and NCC activation [3,4], thus, the Na^+^ is delivered to the distal nephron. The increased Na^+^ in the distal nephron stimulates K^+^ secretion through ROMK due to an electrochemical gradient generated by reabsorption of Na^+^ through ENaC.

These channels are regulated by a group of serine/threonine kinases, WNKs. Mutations in WNK1 and WNK4 genes cause a hereditary disease known as pseudohypoaldosteronism type II (PHAII) characterized with hyperkalemic hypertension [5]. WNK4 is a physiological Cl^−^ sensor that manipulates dietary K^+^ intake [6] and regulates NCC activation through downstream kinases SAPK and OSR1 [7]. Chloride efflux from the cells occurs in K^+^ deficiency, resulting in low intracellular Cl^−^ ([Cl^−^]_i_) that stimulates WNK4 kinase, which phosphorylates SPAK and thus induces NCC phosphorylation [8,9]. The acute K^+^ loading by oral gavage dephosphorylates NCC in wild-type (WT) mice after 30 min oral gavage of K^+^. The decrease in phospho-NCC is not observed in the WNK4-Cl^−^ insensitive knock-in mice. These results indicated that high extracellular K^+^ by increasing [Cl^−^]_i_ inhibits WNK4 and thus inactivates NCC [6]. Interestingly, long-term K^+^ loading still dephosphorylates NCC in the WNK4-Cl^−^ insensitive knock-in mice, indicating that other molecules may be involved in HK-inhibited WNK4 and its downstream signaling [6].

WNK4 also modulates NKCC2 activity. NKCC2 abundance and NKCC2 activity are lower in WNK4*^−/−^* mice than that in controls [10]. Phosphorylation of NKCC2 is regulated by Ste-20 family kinases, including SPAK [11] and OSR1 [12]. SPAK mutant mice have a SPAK-activation deficiency, manifest reduced NKCC2 phosphorylation at T96, and are substantially hypotensive [13]. In addition to T96, NKCC2 overexpressed in the cells is phosphorylated at S91, T100, T105, and S130 by SPAK/OSR1 activation under hypotonic low-chloride conditions. Mutation of T105 or S130 reduces NKCC2 activity by 30–40% [14]. NKCC2 is known to account for approximately 20–25% of Na^+^ reabsorption in the kidney [15], and phosphorylation of T105 and S130 plays the most important role in stimulation of NKCC2 activity. However, S130 phosphorylation has not been detected in the mouse kidney and the mechanism of regulation of phosphorylation of NKCC2 at S130 in vivo is unclear [14].

Na^+^ delivered from K^+^ loading-inhibited NCC and NKCC2 is reabsorbed through ENaC. Hence, K^+^ loading induces ENaC expression and increases the channel activity to prevent Na^+^ loss. ENaC is composed of the α, β, and γ subunits, which are delivered to the apical surface after the synthesis. The activity of ENaC at the apical surface is regulated by proteases, which cleave the α- and γ-ENaC subunits to increase open probability of the channel [16]. An increase in dietary K^+^ intake significantly increased both ENaC and ROMK currents; however, K^+^ loading-induced stimulation of Na^+^ and K^+^ currents was smaller in mice carrying PHAII-mimicking mutations [17]. These results indicate that molecules downstream of WNK4 may be involved in K^+^ loading-regulated ENaC and ROMK activity.

We found that MST3 expression is higher in the medullary thick ascending limb (TAL) than that in the distal convoluted tubules (DCTs) in mice [18], and MST3 protects MDCK cells from physiological hypertonic stress in vitro [19]. To investigate whether MST3 was involved in ion homeostasis, we generated MST3-targeted mutant mice. Since complete knockout of MST3 was not achievable, we reported the phenotype of MST3 hypomorphic mice (referred to as MST3*^−/−^* mice) that manifested enhanced ENaC activity and hypertension [18]. These results indicated that MST3, similar to other Ste20 family members, played an important role in the maintenance of Na^+^ homeostasis. In the present study, we investigated whether MST3 was involved in the regulation of Na^+^ and K^+^ homeostasis in response to K^+^ loading. The expression levels of WNK4, ROMK, BK, ENaC, NCC, and NKCC2 were determined in mice fed the control and HK diets. Plasma Na^+^ and urinary Na^+^/K^+^ excretion were also assayed.

## 2. Results

### 2.1. An Increase in MST3 Levels in Mouse Kidneys with Increasing K^+^ Intake

Since we previously reported that MST3*^−/−^* mice have higher ENaC activity [18], we hypothesized that MST3*^−/−^* mice have higher ability to reabsorb Na^+^ with a low-Na (LNa, 0.04% Na) diet challenge. To preserve Na^+^ in Na^+^ deficiency, WT (MST3*^+/+^*) mice reduced urinary Na^+^ excretion from 782.62 ± 152.57 to 90.89 ± 61.78 μmol/day (Figure 1A,B) and reduced urine volume from 5345.00 ± 861.55 to 3748.67 ± 789.03 μL/day (Figure 1C,D). MST3*^−/−^* mice also preserved Na^+^ through a reduction in urinary Na^+^ excretion from 589.13 ± 90.61 to 61.78 ± 22.26 μmol/day (Figure 1A,B) and urine volume from 3958.00 ± 1047.19 to 2729.31 ± 861.14 μL/day (Figure 1C,D). The urinary Na^+^ excretion ratio of low Na (LNa) to control diets was 12.11 ± 3.8% in WT mice and 10.42 ± 2.7% in MST3*^−/−^* mice. The urine volume ratio of LNa to control diets was 70.58 ± 13.19% in WT mice and 67.55 ± 12.91% in MST3*^−/−^* mice. There were no differences between WT and MST3*^−/−^* mice in Na^+^ reabsorption in Na^+^ deficiency. However, we previously reported that MST3 protein level is upregulated in WT mice after high-salt (HS) intake (8% Na, 1.1% K) [19]. The HS diet-fed mice intake two-fold higher levels of water and chow than those animals fed the control diets (Table 1), indicating that the animals intake higher levels of both Na^+^ and K^+^. To investigate the effects of Na^+^ or K^+^ separately, we fed mice with increasing Na^+^ and increasing K^+^ by adding 1% NaCl and 1% KCl in drinking water to determine the effects of Na^+^ and K^+^ on MST3. The MST3 expression was similar in kidney to that of control, LNa or 1.43% Na diet (0.43% Na in chow with additional 1% NaCl in drinking water) (Figure 2A). Interestingly, 2% KCl (1% K in chow with additional 1% KCl in drinking water) stimulated MST3 expression (Figure 2B). To determine the effect of K^+^-loading on MST3 function in the kidney, we fed WT and MST3*^−/−^* mice 2% KCl diets. The 2% KCl diets stimulated an approximately 1.6-fold increase in MST3 expression in WT mice (Figure 2C, lanes 4–6). MST3*^−/−^* mice consistently expressed a lower level of MST3 than that in WT mice (Figure 2C, lanes 7–9); however, we observed only a 1.1-fold increase in MST3 expression in 2% KCl diet-fed MST3*^−/−^* mice (Figure 2C, lanes 10–12).

### 2.2. Reduction of Diuresis, Kaliuresis, and Natriuresis in MST3*^−/−^* Mice Fed 2% KCl Diets

Consistent with the notion that K^+^ loading causes diuresis and kaliuresis [20], K^+^ loading induced a rapid diuresis in WT mice (urine volume from 4812.22 ± 695.51 to 5700.80 ± 792.40 μL) on day 1 of K^+^ loading and remained at 6813.83 ± 2229.64 μL during days 2–6 of K^+^ loading. However, the urine volume of MST3*^−/−^* mice was increased only from 3884.58 ± 695.51 to 5875.00 ± 993.79 μL on day 1 of K^+^ loading and was reduced to 3652.50 ± 893.94 μL on subsequent days of K^+^ loading (Figure 3A,B). A kaliuresis was also observed from 1082.92 ± 136.58 to 1692.90 ± 136.58 μmol on day 1 of K^+^ loading and remained at 1903.83 ± 435.69 μmol/d in WT mice (Figure 3C,D). These results indicated that in response to increasing K^+^ intake, the kidneys normally excreted approximately 1.75-fold of K^+^, which was between 90 to 95% of the daily 2-fold increase of K^+^ intake (Figure 3D); however, urinary K^+^ excretion in MST3*^−/−^* mice was substantially increased from 948.49 ± 105.69 to 1873.12 ± 370.63 μL on day 1 of K^+^ loading and was then slightly increased to 1230.35 ± 205.69 μL (Figure 3C,D). Comparison of the fold change of the urine volume on K^+^ loading to the control diets indicated an approximately 1.4-fold increase after K^+^ loading in WT mice. The urine volume in K^+^ loading-treated MST3*^−/−^* mice was only 0.8-fold of that in control diet-fed mice, significantly less than the 1.4-fold increase in WT mice (*p =* 1.8 × 10^-5^) (Figure 3). When comparing the fold change of K^+^ secretion on K^+^ loading, only a 1.29-fold increase was detected in MST3*^−/−^* mice, which was significantly less than a 1.75-fold increase in WT mice (*p =* 4.0 × 10^-6^) (Figure 3D), indicating that MST3*^−/−^* mice exhibited reduced diuresis and kaliuresis than that in WT mice on K^+^ loading. 

The urinary Na^+^ excretion in WT mice was slightly reduced from 763.10 ± 87.68 to 680.59 ± 44.45 μmol on day 1 of K^+^ loading and returned back to 712.87 ± 102.34 μmol/d on subsequent days of K^+^ loading, indicating that Na^+^ was maintained at homeostasis. In contrast, urinary Na^+^ excretion was significantly decreased in MST3*^−/−^* mice from 663.74 ± 79.60 to 532.68 ± 71.95 μmol after K^+^ loading (Figure 3E,F) and reduced to approximately 0.8-fold of that in mice fed control diet; these levels were significantly less than those in WT mice (0.93-fold change, *p =* 0.0006) (Figure 3F). These results indicated that MST3*^−/−^* mice reabsorbed higher amounts of Na^+^ and water than those in WT mice on 2% KCl diets. This increase in Na^+^ reabsorption was associated with an increase in the plasma [Na^+^] (in mM, 154.67 in MST3*^−/−^* mice vs. 152.5 in MST3*^+/+^* mice) (Table 2). Overall, the Na^+^- and flow-dependent K^+^ secretion was inhibited in MST3*^−/−^* mice. After K^+^ loading, systolic blood pressure (SBP) of WT mice was 119 ± 10 mm Hg, which was similar to the SBP in mice fed the control diet; however, SBP of MST3*^−/−^* mice was 131 ± 9 mm Hg (Table 2). These results suggested that only a 1.1-fold increase in MST3 in MST3*^−/−^* mice fed the 2% KCl diets was insufficient to excrete Na^+^ and K^+^, causing elevated SBP in MST3*^−/−^* mice.

### 2.3. WNK4 and WNK4-Regulated Channels in 2% KCl Diet-Fed Mice

WNK4 plays an important role in modulating renal K^+^ secretion and Na^+^ absorption. We found that 2% KCl diets induced increased WNK4 expression in MST3*^−/−^* mice (Figure 4A), indicating that MST3 might be involved in WNK4 regulation. WNK4 have been shown to inhibit ROMK activity by stimulating clathrin-mediated endocytosis [21] and inhibiting maxi-K by a kinase-dependent mechanism [22]. Figure 4B showed that BK and ROMK were increased after K^+^ loading in WT mice. In MST3*^−/−^* mice, there was no obvious difference in BK expression; however, ROMK was not induced after K^+^ loading, which may cause reduced kaliuresis. 

Consistent with the notion that K^+^ loading induced ENaC γ-cleavage [6], 2% KCl diets induced increased cleaved γ-ENaC expression in WT mice, indicating that 2% KCl diets increased ENaC activity. Compared with WT mice, MST3*^−/−^* mice had a higher cleaved γ-ENaC on control diets and higher full-length ENaC on 2% KCl diets (Figure 4C). IHC results showed that feeding the 2% KCl diets slightly increased γ-ENaC expression at the apical plasma membrane of the DCT2/CNT in WT mice compared with that in mice fed the control diets (Figure 4D, c vs. a). Consistent with our previous report, MST3*^−/−^* mice exhibited higher ENaC expression at the apical plasma membrane of the DCT2/CNT. A higher intensity of ENaC staining was observed on 2% KCl diets (Figure 4D, g vs. e). The MST3 expression showed that higher levels of MST3 protein were present in the cytosol of the DCT2/CNT in WT mice on 2% KCl diets compared to that in WT mice on control diets (Figure 4D, d vs. b). Lower levels of MST3 were observed in MST3*^−/−^* mice on 2% KCl diets (Figure 4D, h). 

Isoforms A, B, and F of NKCC2 are estimated to account for 20–25% of all renal Na^+^ reabsorption. The NKCC2-F isoform mainly located in the inner medullary TAL accounts for 70% of NKCC2 expression [23]. Since MST3 is primarily localized in the inner medullary TAL, we determined whether MST3 is involved in K^+^ loading-mediated NKCC2 phosphorylation. K^+^ loading inhibited nonglycosylated NKCC2 phosphorylation at S130 in WT mice; however, the level of nonglycosylated and glycosylated phospho-S130-NKCC2 was increased in MST3*^−/−^* mice on 2% KCl diets (Figure 4E). These results indicated that MST3 inhibited NKCC2F phosphorylation at S130. IHC results showed that both MST3 (Figure 4F, G, b) and NKCC2 (Figure 4F, G, a) are mainly expressed at the apical membrane of the inner medullary TAL in control diet-fed WT mice. A 2% KCl diet induced MST3 expression ((Figure 4F, G, d) in the cytosol of the inner medullary TAL. NKCC2 was present at the subapical membrane of the inner medullary TAL (Figure 4F, G, c vs. a). This pattern is more clearly observed in an enlarged image (Figure 4G). However, in 2% KCl diet-fed MST3*^−/−^* mice, low levels of MST3 and higher levels of NKCC2 were still present at the apical membrane of the inner medullary TAL ((Figure 4F, 4G, g and h). These results indicated that MST3 inhibited medullary NKCC2 expression at the apical membrane of the medullary TAL in mice fed the 2% KCl diets. 

The 2% KCl diets reduced the level of NCC in MST3*^+/+^* and MST3*^−/−^* mice (Figure 4H, lower panel), indicating that K^+^ loading-inhibited NCC expression promoted K^+^ secretion. However, there were no differences in phospho-NCC levels in MST3*^+/+^* and MST3*^−/−^* mice fed the 2% KCl diets (Figure 4H, upper panel). Additionally, there were no differences in the NCC distribution in the DCT1 in mice fed the control and 2% KCl diets (Figure 4I). These results indicated that MST3 may have a small or no effect on K^+^ loading-mediated inhibition of NCC activity.

## 3. Discussion

K^+^ preferentially leaves the cells through K^+^ channels, such as ROMK and BK, at the apical membrane of the DCT2/CNT. This process is driven by an electrochemical gradient generated by reabsorption of Na^+^ through ENaC to induce a K^+^-secreting state. Na^+^ delivered to DCT2/CNT is due to K^+^ loading-induced NCC and NKCC2 inhibition; thus, K^+^ loading-inhibited NCC and NKCC2 and K^+^ loading-induced ENaC activation needs to be strictly regulated to maintain Na^+^ homeostasis (Figure 5). We found that the 2% KCl diets induced higher MST3 expression in WT mice than that in MST3*^−/−^* mice. MST3*^−/−^* mice with reduced MST3 expression had higher WNK4 expression, which might be involved in ENaC activity and NKCC2 phosphorylation at S130. These results indicated that MST3*^−/−^* mice reabsorbed more Na^+^ at TAL, thus reducing K^+^ secretion. In DCT2/CNT, MST3*^−/−^* mice had higher ENaC activity than that in WT mice, indicating that MST3*^−/−^* mice could not inhibit ENaC activation to prevent ENaC overactivation. Overall, MST3*^−/−^* mice reabsorbed more Na^+^ and K^+^ than did WT mice on HK diets. Our results indicate that MST3 functions to maintain Na^+^ and K^+^ homeostasis in mice on 2% KCl diets in vivo.

K^+^ loading induces an increase in the circulating plasma levels of aldosterone to stimulate K^+^ secretion, and then aldosterone has a smaller increase for longer periods [24]. In addition, HK-induced K^+^ secretion is dependent on Na^+^ delivery and flow, resulting from inhibition of Na^+^ reabsorption in the TAL and DCT1. The resultant Na^+^ delivery and flow along with increased aldosterone facilitate renal K^+^ excretion through ROMK and BK channels [25,26,27]. Our results indicated that both WT and MST3*^−/−^* mice exhibited apid diuresis and kaliuresis on the 1st day K^+^ loading; however, MST3*^−/−^* mice could not continually increase K^+^ secretion on the subsequent days of HK challenge (Figure 3). We suggested that aldosterone might be involved in K^+^ secretion at the beginning of the K^+^ challenge, and then MST3 plays a role in Na^+^-dependent and flow-dependent K^+^ secretion. 

Most studies used 5% K^+^ in an HK diet; feeding this diet decreased the abundance of both NCC and phospho-NCC [6,8]. However, 5% K^+^ is unphysiological. Modest changes in dietary K^+^ affect plasma [K^+^] and NCC in a graded manner [8]. We fed mice with modest changes of K^+^ by increasing K^+^ from 1% (1% in chow) to 2% K^+^ (1% in chow and 1% in drinking water). An approximately 2-fold increase of urinary K^+^ was excreted in response to increased K^+^ intake (from 1% to 2%). The abundance of NCC was reduced in both MST3^+/+^ and MST3*^−/−^* mice, indicating that NCC was inhibited, thus promoting K^+^ secretion (Figure 4H, lower panel); however, phospho-NCC was not obviously inhibited (Figure 4H, upper panel), which may be due to the modest K^+^ challenge in mice in the present study. The results of IHC analysis showed a lack of differences in NCC distribution in MST3*^+/+^* and MST3*^−/−^* mice (Figure 4I). These results suggested that the lowest MST3 expression in the DCT1 was not involved in 2% KCl loading-inhibited NCC activation in vivo.

WNK4 kinase activity is regulated by different mechanisms to transduce signals to downstream molecules. The KLHL3/CUL3 ubiquitin ligase complex degrades WNK4. In PHAII, the loss of interaction between KLHL3 and WNK4 increases levels of WNK4 [28]. Phosphorylation of WNK4 by PKC and PKA regulate the WNK4′s activity and downstream signaling [29]. Protein phosphatase 1 binds to WNK4 and modulates the inhibitory effect of WNK4 on ROMK [30], and activation of protein phosphatases (PPs) may mediate NCC dephosphorylation in response to high extracellular K^+^ [31]. In addition, WNK4 is a Cl^−^ sensor. WNK4 regulates WNK4 activity by binding to Cl^−^. A WNK Cl^−^-sensing mechanism explains WNK-mediated regulation of NCC/NKCC2 by diets with various levels of K^+^. At high intracellular chloride concentrations ([Cl^−^]_i_), chloride ions binds to WNK4, thus inhibiting WNK4 activity. The acute K^+^ loading-dephosphorylated NCC was not observed in the WNK4-Cl^−^ insensitive knock-in mice, indicating that high extracellular K^+^ by increasing [Cl^−^]_i_ inhibits WNK4 and thus inactivates NCC [6]. However, the long-term K^+^ loading still dephosphorylates NCC in the WNK4-Cl^−^ insensitive knock-in mice, indicating that another mechanism was involved in HK-inhibited WNK4 and its downstream signaling. MST3*^−/−^* mice exhibited higher WNK4 expression (Figure 4A), higher ENaC activity (Figure 4C,D), and higher p-NKCC2 (Figure 4E,G) on 2% KCl diets for 16 days, indicating that MST3 was involved in WNK4 and its downstream signals. We have previously reported that MST3 was phosphorylated at the tyrosine residues. Tyrosine phosphorylation of MST3 may create a docking site for molecules involved in diverse signaling pathways [32]. We demonstrated that MST3 inhibited protein tyrosine phosphatase activity to inhibit cell migration through paxillin regulation [33]. Involvement of MST3 in phosphatase activity, ubiquitination, or WNK4 phosphorylation, which regulates ENaC activity and NKCC2 phosphorylation in the case of long-term K^+^ loading, requires additional investigation. 

Dietary potassium inhibits NCC- and NKCC2-mediated Na^+^ reabsorption and shifts Na^+^ downstream for reabsorption by ENaC, which can drive K^+^ secretion and prevent Na^+^ loss. This study reports that increased K^+^ intake stimulates MST3 expression to inhibit Na^+^ reabsorption. This effect is mediated by inhibition of NKCC2 and ENaC; inhibition of NKCC2 inhibits Na^+^ reabsorption and promotes K^+^ secretion; inhibition of ENaC does not increase Na^+^ reabsorption, thus maintaining Na^+^ homeostasis (Figure 5).

## 4. Methods

### 4.1. Animals

C57BL/6 male mice in Table 1 were 12-weeks-old and housed in metabolic cages and allowed ad libitum access to food and water for 4 weeks [19]; the animals were divided into two groups: the control diet group (diets: 0.43% Na and 1.1% K (*w/w*)) and high-salt (HS) diet group (diets: 8% Na and 1.1% K (*w/w*); TestDiet, St. Louis, MO, USA). MST3*^+/+^* (WT) and MST3 hypomorphic mutant (designated MST3*^−/−^*) mice were obtained as reported previously [18]. To investigate the effect of an increase in Na^+^ or K^+^ separately, we added additional 1% NaCl and 1% KCl into drinking water to make sure that the mice took in additional Na^+^ and K^+^. The mice were fed control, low Na (LNa), 1.43% Na (0.43% Na in chow with additional 1% NaCl in drinking water), and 2% KCl (1% K in chow with additional 1% KCl in drinking water) diet. Male WT and MST3*^−/−^* mice (8–12-weeks-old) were allowed ad libitum access to food and water. The mice were kept in metabolic cages, fed the control diet for the first 3 days and then challenged with LNa and HK diets for the following 6 days. Urine was collected during this period. Then, the animals were moved to the mouse cages and exposed to the corresponding challenge diet for the next 10 days. 

### 4.2. Immunohistochemistry

The procedures have been described previously in detail. Briefly, serial sections of mouse kidneys were deparaffinized in xylene before rehydration in a graded series of ethanol and then the sample was incubated in a buffer (1 mM Tris in PBS, pH 8.2) at 100 °C for 20 min for antigens retrieval. Serial sections were incubated with antibodies against MST3 (1:500, a gift from Dr. Ming-Derg Lai, Taiwan), γ-ENaC (1:200, cat. no. 13943-1-AP; Proteintech, IL, USA), NCC (1:8000, cat. no. ab3553; Millipore, MA, USA), and NKCC2 (1:200, cat. no. AF2850; R&D Systems, MN, USA). Specificity of the anti-MST3 antibody was confirmed as previously report [18,33,34]. The secondary antibodies (1:1000, cat. no. 111-035-144; Jackson ImmunoResearch, PA, USA) were incubated at room temperature for 1 h with subsequent 3,3′-diaminobenzidine (DAB) (DAKO, Denmark, Hilden, Germany) staining.

### 4.3. Immunoblotting

Kidneys harvested from mice were homogenized using a PT 2100 Polytron homogenizer in ice-cold lysis solution containing 50 mM HEPES, pH 7.2, 150 mM NaCl, 1% Triton X-100, and protease inhibitors. The lysate from the kidney was first centrifuged at 100,000× *g* for 30 min at 4 °C. Samples from the supernatant were resolved by SDS-PAGE and transferred to NC paper. Western blot analysis was performed as reported previously [18] using primary antibodies against MST3 (1:1000), WNK4 (1:1000, cat. no. 22326-1-AP; Proteintech,, IL, USA), BKα (1:500, cat. no. APC-151; Alomone labs, Jerusalem, IL), ROMK (1:500, cat. no. APC-001; Alomone labs, Jerusalem, IL), γ-ENaC (1:1000), NCC (1:500), NKCC2 (1:1000), phospho-T53-NCC (1:500, cat. no. ab254039; Abcam, Cambridge, UK), and phospho-S130-NKCC2 (1:500, a gift from Dr. Dario Alessi, UK).

### 4.4. Measurement of Blood Pressure, Serum Na^+^, and Urinary Concentrations of Na^+^ and K^+^ and Statistical Analysis

The steady-state SBP (systolic blood pressure) of restrained conscious mice was measured by a programmable tail-cuff sphygmomanometer (MK-2000ST, Muromachi, Tokyo, JP). SBP was initially estimated by inflating the cuff at approximately 25 mm Hg/sec. SBP was accurately determined during cuff deflation at approximately 4–5 mm Hg/sec. SBP was defined as blood pressure (BP) corresponding to the reappearance of the pulse. Blood samples were obtained via cheek pouch bleeding. Urine was collected from the urine collection tubes of the metabolic cages every 24 h for 9 days. The concentrations of Na^+^ and K^+^ were measured using an Advia 1800 chemistry system (Siemens). The levels of Na^+^ and K^+^ in urine and urine volume are shown as the mean ± SD. The statistical analysis was performed using Microsoft Excel 2013 by one-way analysis of variance (ANOVA) followed by Bonferroni post hoc tests. *p*-values * <0.05, ** <0.005, and *** <0.0005 were considered significant. The mean values of the animals fed the control diet were averaged for the first 3 days of the treatment.

## Figures and Tables

**Figure 1 ijms-22-00999-f001:**
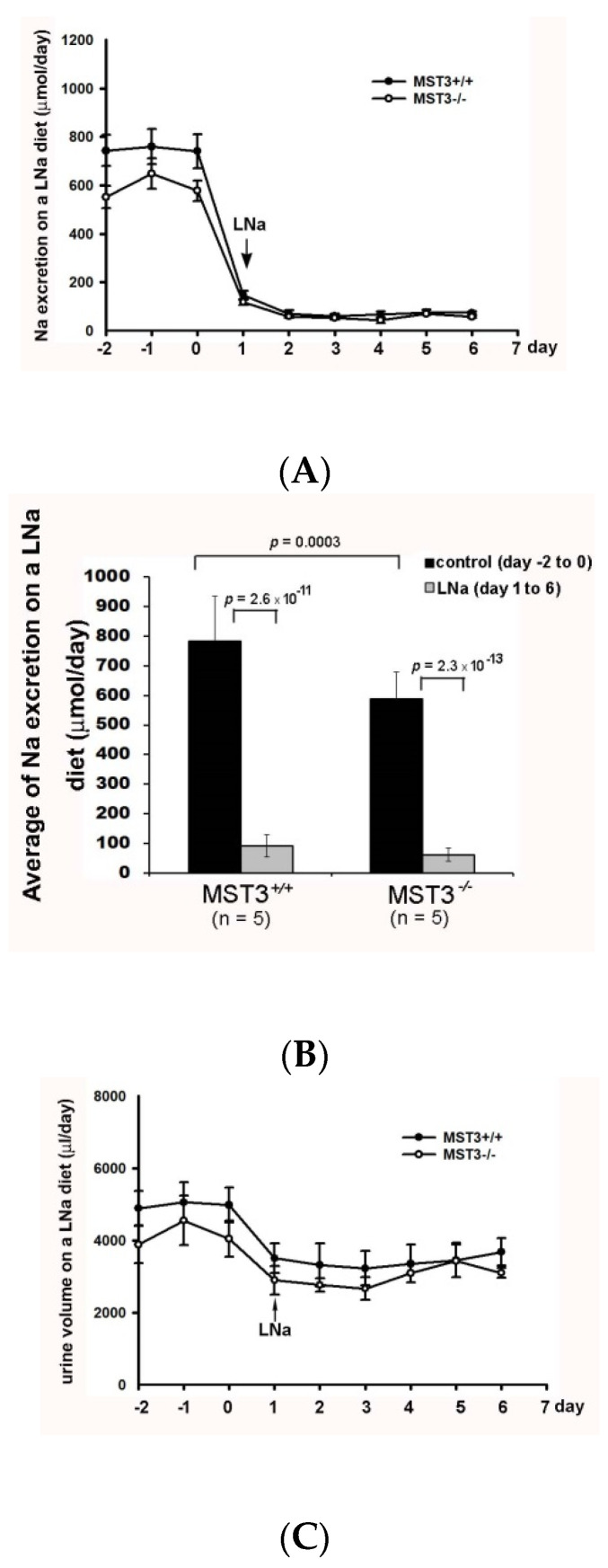
Urinary Na^+^ excretion and urine volume on a low Na (LNa) diet. Wild-type (WT) (MST*^+/+^*) (●) and MST3*^−/−^* (○) mice were fed the control diet for 3 days; then, the diet was changed to the LNa diet for an additional 16 days. The average group values were used to generate the graphs, and the error bars correspond to SE. Urinary Na^+^ excretion (**A**) and urine volume (**C**) were recorded over 9 days. The bar graphs (**B**,**D**) show the average of the data in (**A**,**C**), respectively.

**Figure 2 ijms-22-00999-f002:**
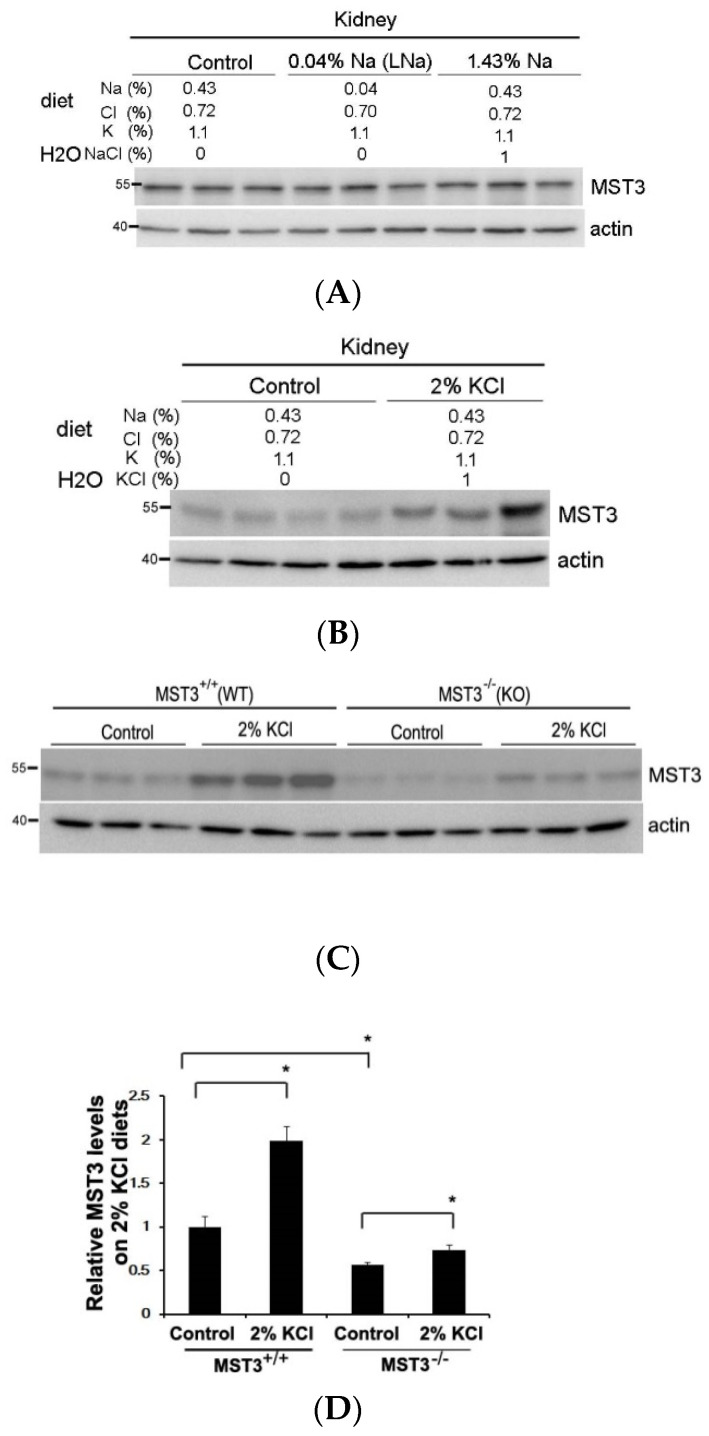
A 2% KCl diet stimulates MST3 expression. MST3 protein expression was detected in WT mice fed the (**A**) control (n = 3 or 4 animals), low Na (LNa), 1.43% Na or (**B**) 2% KCl diets for 16 days (n = 3 animals/group). The sodium, potassium, chloride and water contents in the diet are indicated. (**C**) MST3 protein expression was detected in WT and MST3*^−/−^* mice fed the control and 2% KCl diets (n = 3 animals/group). (**D**) The bar graph shows quantification of MST3 expression in (**C**). * *p* < 0.05 vs. the control group.

**Figure 3 ijms-22-00999-f003:**
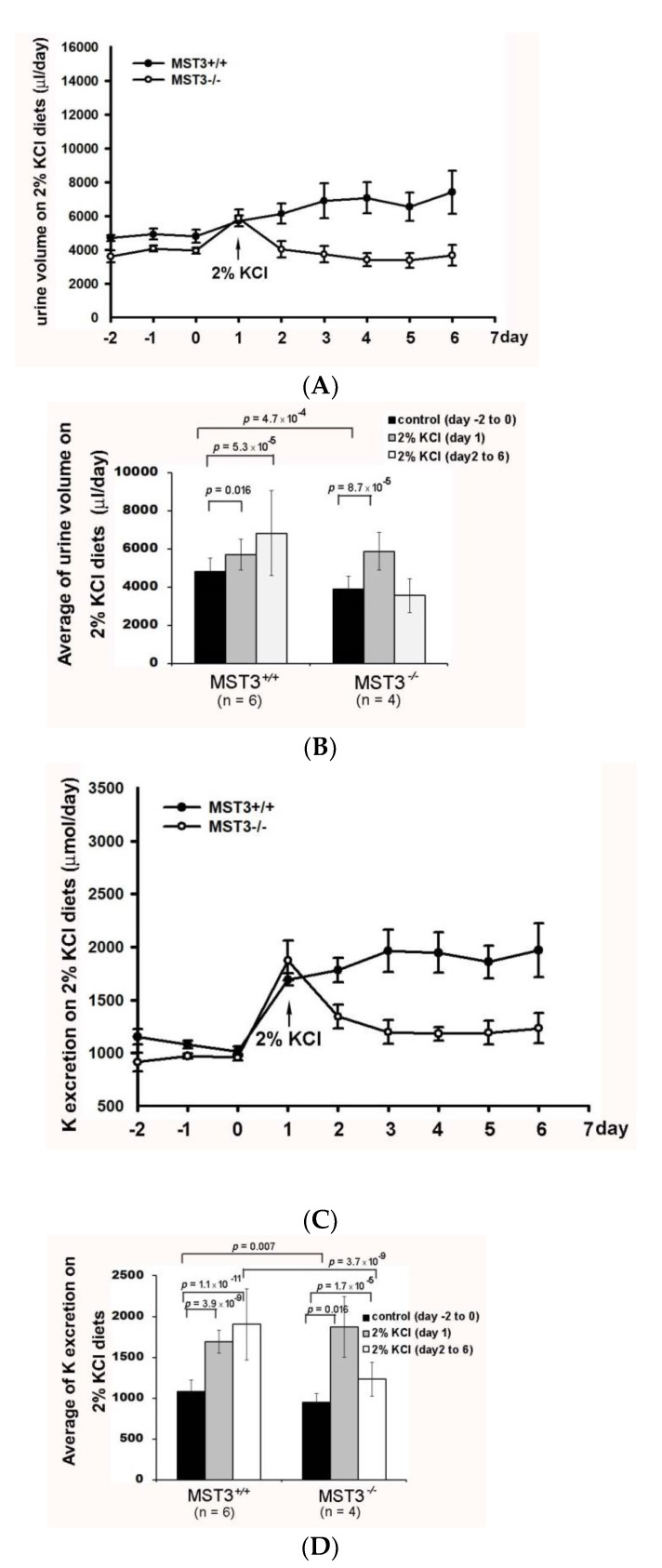
Reduction of diuresis, kaliuresis, and natriuresis in MST3*^−/−^* mice fed 2% KCl diets. MST*^+/+^* (●) and MST3*^−/−^* (○) mice were fed a control diet for 3 days; then, the diet was changed to the 2% KCl diet for an additional 16 days. The urine volume (**A**), urinary K^+^ (**C**), and urinary Na^+^ (**E**) excretion were recorded over 9 days. The average group values were used to generate the graphs, and the error bars correspond to SE. The bar graphs (**B**,**D**,**F**) show the average of the data in (**A**,**C**,**E**) respectively.

**Figure 4 ijms-22-00999-f004:**
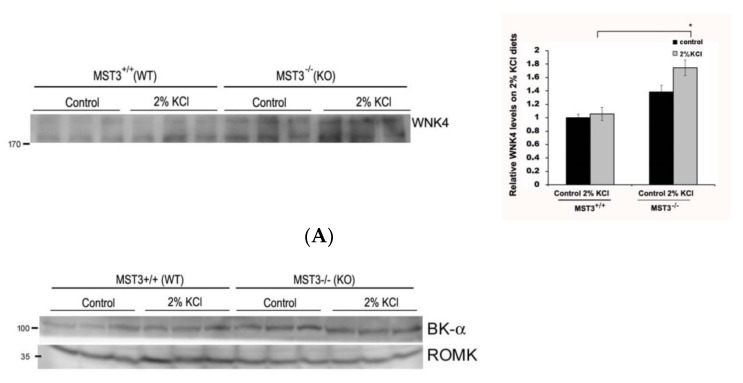
WNK4 and WNK4-regulated channels in mice fed 2% KCl diets. Western blot analysis of (**A**) WNK4, (**B**) BK, and ROMK, (**C**) γ-ENaC, (**E**) NKCC2, and p-NKCC2, and (**H**) NCC and p-NCC in the kidney of MST*^+/+^* and MST3*^−/−^* mice 8 weeks after the treatment with the control and 2% KCl diets for 16 days (*n* = 3 animals/group). The bar graph shows quantification of Western blot relative to the levels in the control group of WT mice. * *p* < 0.05 vs. the control group. Serial sections of the kidney of WT and MST3*^−/−^* mice fed the control and 2% KCl diets were stained for γ-ENaC (**D**), NKCC2 (**F**,**G**, outlined images of **F** were enlarged using a 100× objective.), NCC (**I**), and MST3. G, glomerular; D1, early distal convoluted tubule; D2, late distal convoluted tubule; CN, connecting tubule.

**Figure 5 ijms-22-00999-f005:**
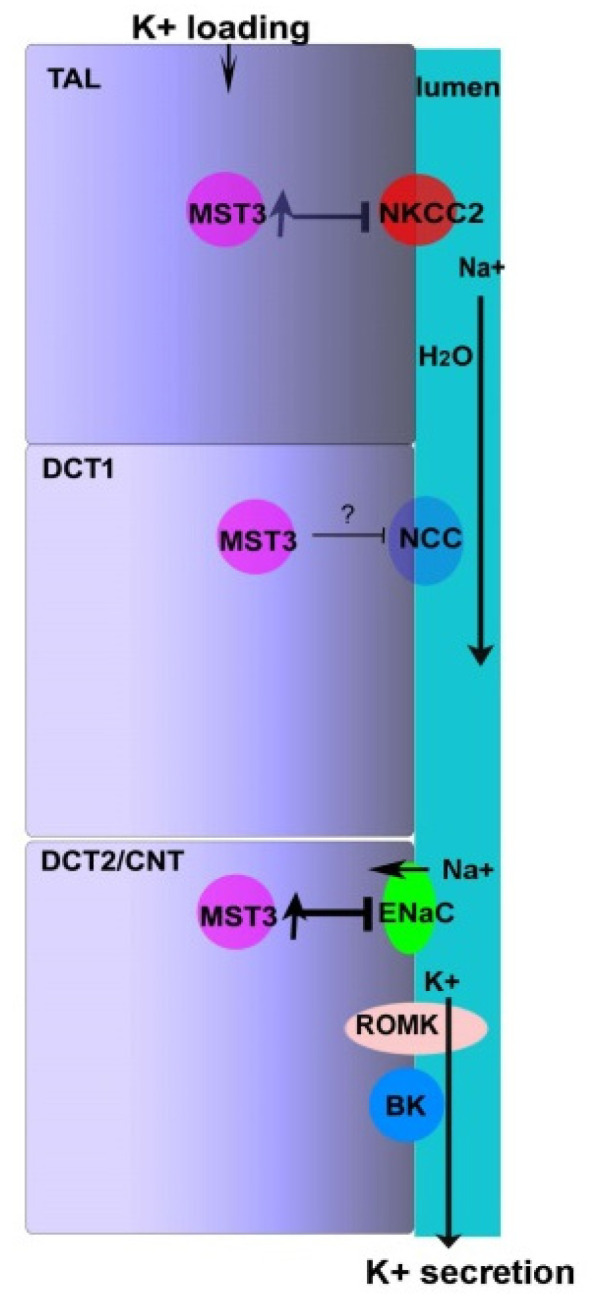
Hypothetical model of MST3 function on 2% KCl diets. NKCC2 and NCC in TAL and DCT1, respectively, are inhibited upon an increase in K^+^ intake, thus, delivering Na^+^ and water to the DCT2/CNT. Na^+^ in the DCT2/CNT is reabsorbed through ENaC, resulting in K^+^ secretion due to an electrochemical gradient. Reabsorption of Na^+^ through ENaC at the DCT2/CNT prevents Na^+^ loss. An increase in K^+^ intake stimulates MST3 expression, which inhibits NKCC2 and ENaC expression at the apical membrane of the nephron, thus preventing excessive absorption of Na^+^ and maintaining Na^+^ homeostasis.

**Table 1 ijms-22-00999-t001:** The values of weight, water intake, and food intake in 8-week-old C57Bl/6 male mice fed control or high salt diets.

Diet	Control Diet (0.43% Na, 1.1% K) (*n* = 5)	High Salt (8% Na, 1.1% K) (*n* = 5)
Weight, g	19.55 ± 2.12	21.94 ± 1.52
Water intake, mL	4.96 ± 0.52	8.62 ± 1.01
Food intake, g	3.17 ± 0.43	8.82 ± 1.89

**Table 2 ijms-22-00999-t002:** Blood pressure (BP) in WT and MST3*^−/−^* mice on control diets for 3 days and challenged with 2% KCl diets. * *p* < 0.05 vs. the WT group.

	Value		
	16 Days	MST3*^+/+^* (WT)	MST3 ^−/−^
Plasma [Na^+^] (mmol/L)	control diet	152.33 ± 0.52	153.4 ± 0.89 *
	2% KCl diets	152.5 ± 0.84	154.67 ± 1.53 *
Blood pressure (mmHg)	control diet	117 ± 9	130 ± 13 *
	2% KCl diets	118 ± 10	131 ± 9 *

## Data Availability

Not applicable.

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
