# Peer review of "MST3 Involvement in Na+ and K+ Homeostasis with Increasing Dietary Potassium Intake"

_ijms, 2021, doi:10.3390/ijms22030999_

Round 1

Reviewer 1 Report

Chan et al. present a manuscript showing their results on the function of mammalian Ste20-like protein kinase 3 (MST3/STK24) for the adaptation of sodium and potassium homeostasis in the nephron. The authors used mice with a hypomorphic mutation on MST3 and exposed them to high potassium diets. The authors demonstrated that high potassium diets induce MST3 expression that, in turn, increased the phosphorylation and the inhibition of the sodium-potassium-chloride cotransporter, the amount of the epithelial sodium channel, and the recapture of sodium from tubular fluids, which drive the excretion of potassium. The authors wrote the paper with a clear English style. The problem is interesting and original, the experiments adequately designed, the results compelling and appropriately analyzed, and the experiments well designed. Nonetheless, phosphorylated sodium-potassium-chloride cotransporter blots are non-specific, which prevents to conclude that there is an increase of phosphorylation on serine 130. The none-specific phosphorylation blots preclude the publication of the manuscript in its present form. Therefore, this reviewer encourages the authors to present a corrected manuscript showing: compelling phosphorylation blots, new phosphorylation experiments using other techniques, or a discussion on the weaknesses of their phosphorylation data.

Specific comments

  1. In which conditions are animals (including humans) exposed to a low potassium diet? Authors may consider adding this information to boost the interest of readers.
  2. Why did the authors choose the hypomorphic MST3 mutation and no other? Was it designed on purpose or detected incidentally? The addition of this information would help to understand the experimental design. 
  3. Which data or information support the proposition that other members of the Ste20 family “may be involved in HK-induced NCC and NKCC2 dephosphorylation”? Are these analogous kinases? Do similar results operate in other systems or conditions? The authors may consider adding this information to strengthen their argument.
  4. “An increase in MST3 levels in mice fed the HK diets.” The authors may clarify the figure 1 legend title. Example: “HK diet increases MST3”.
  5. The graph indicates that the diet changed to high potassium on day one, while the text specifies that it was at zero time. Authors may consider adjusting the arrow to zero time.
  6. The authors must substitute the asterisks on graphs with the real p values or write them on results or the figure legends.
  7. “The HS diet-fed mice intake two-fold higher levels of water and chow than those animals fed the control diets (Table I)”. Precise the phrase, add the word “around” or any other synonym before “two-fold”.
  8. Clarify figure legend 2 and the corresponding title of results as suggested above in 4.
  9. Authors must explain or change the procedure to calculate “fold of increase” in the manuscript (Figure 1I, lines 218-224, 231 to 232, and so on). Authors may consider re-drawing the graph Indicating the control value (1) with a line, and the increases and decreases of the ratios with bars going up and down the line, respectively. That kind of graph would show clearly the real, amount of change.
  10. The bands of the blots in Figure 4D F are somewhat weak and, therefore, not compelling. Authors must provide clear blots and their corresponding analysis, if possible. Otherwise, they must strengthen their immunohistochemistry results by other means (for example: testing another antibody or using immunofluorescence confocal microscopy), or discuss the limitations of these results. It is also feasible to immuno-precipitate MST3 from kidney extracts, and perform anti phosphor serine blots to measure the phosphorylation degree more clearly. 
  11. Figures 4 C, F, G, and J: the authors may obtain quantitative data of the images and show whether the results correspond to the ones obtained from blots.

Reviewer 2 Report

Authors investigated to Na+/K+ homeostasis with a high K+ diet. In MST-/- mice, HK diet cannot inhibit NKCC2 activity at TAL and activate ENaC at DCT. I have some comments and questions below to improve this manuscript.

  1. Please show blood pressure of MST3-/- mice with HK diet. In Liddle syndrome, activated ENaC shows hypertension. I would like to know whether activated ENaC affect clinical features in MST3-/- mice.
  2. Authors should investigate to quantitative analysis on plasma K+, ROMK and BK to identify K+ homeostasis.
  3. Authors should investigate to the mechanism of ENaC activation. For example, quantitative analysis on alpha-ENaC, Sgk1 and Nedd4-2. They are associated with aldosterone dependent ENaC activation. In addition, ubiquitination is most important to regulate ENaC. However, some researchers reported that phosphorylation affected ENaC activation via WNK4. Authors should show quantitative analysis on WNK4 in MST3-/- mice with HK diet.
  4. Abstract is so narrative and there is no reference about MST3 in Introduction. Please describe MST3 features in Introduction.
  5. Please check again author’s analysis. I cannot identify the significance in some figures.

Author Response

Dear reviewer, 

We answer your question and revised the manuscript in the attached file.

Thank you very much.

Sincerely,

Lu

Reviewer 3 Report

Overall, this is very well conducted study showing MST3 involvement in Na+
and K+ homeostasis on a high-potassium diet.

There are several comments to improve manuscript.

The definition of high potassium diet needs to be more details and better clarified. 

The investigators should also provide the insights of future needed studies after this study, and what is the implications of this study.

Furthermore, additional discussions on related work on this topic need improvement.

Author Response

(The authors gave the same response as above.)

Round 2

Reviewer 1 Report

The manuscript has greatly improved; it now contains new and original knowledge worth publishing.

Author Response

Thank you for your comments.

Reviewer 2 Report

Authors answered my questions and made theoretical achievements.

I would like to ask authors some questions.

  1. Authors made some figures “Average” and “Fold change” in Fig.1 B-C, Fig.1 E-F, Fig.3 B-C, Fig.3 E-F and Fig.3 H-I. I think only “Average” is enough for these figures.
  2. In Fig.4H, the scale of the vertical axis becomes gamma-ENaC. Is this pNCC?
  3. I found some mistakes in lane 200, 234, 332 and 561. For example, MST-/. MST-/- and Na+. Please correct to MST-/- and Na+.

Author Response

Thank you for your comments.

We have revised the manuscript according to your commnets.

Reviewer 3 Report

It appears that all comments have been appropriately responded to.  I have no further comments and recommend publication.

Author Response

Thank you for your comments.